# Single Column Model Simulations of Icing Conditions in Northern Sweden: Sensitivity to Surface Model Land Use Representation

**Erik Janzon** [1,*] **, Heiner Körnich** [2] **, Johan Arnqvist** [1] **and Anna Rutgersson** [1]

[1] Department of Earth Sciences, Uppsala Universitet, 75236 Uppsala, Sweden;
johan.arnqvist@geo.uu.se (J.A.); anna.rutgersson@met.uu.se (A.R.)
[2] Swedish Meteorological and Hydrological Institute, 60176 Norrköping, Sweden; heiner.kornich@smhi.se
* Correspondence: erik.janzon@geo.uu.se

**Abstract:** In-cloud ice mass accretion on wind turbines is a common challenge that is faced by energy companies operating in cold climates. On-shore wind farms in Scandinavia are often located in regions near patches of forest, the heterogeneity length scales of which are often less than the resolution of many numerical weather prediction (NWP) models. The representation of these forests—including the cloud water response to surface roughness and albedo effects that are related to them—must therefore be parameterized in NWP models used as meteorological input in ice prediction systems, resulting in an uncertainty that is poorly understood and, to the present date, not quantified. The sensitivity of ice accretion forecasts to the subgrid representation of forests is examined in this study. A single column version of the HARMONIE-AROME three-dimensional (3D) NWP model is used to determine the sensitivity of the forecast of ice accretion on wind turbines to the subgrid forest fraction. Single column simulations of a variety of icing cases at a location in northern Sweden were examined in order to investigate the impact of vegetation cover on ice accretion in varying levels of solar insolation and wind magnitudes. In mid-winter cases, the wind speed response to surface roughness was the primary driver of the vegetation effect on ice accretion. In autumn cases, the cloud water response to surface albedo effects plays a secondary role in the impact of in-cloud ice accretion, with the wind response to surface roughness remaining the primary driver for the surface vegetation impact on icing. Two different surface boundary layer (SBL) forest canopy subgrid parameterizations were tested in this study that feature different methods for calculating near-surface profiles of wind, temperature, and moisture, with the ice mass accretion again following the wind response to surface vegetation between both of these schemes.

**Keywords:** wind energy; heterogeneous land use; icing; cold climate; forests

## 1. Introduction

In the past several decades, commercial wind power has grown from humble beginnings in small research wind farms to an important energy resource worldwide. Utilities, independent power producers, and investors have especially taken advantage of regions in cold climates, such as those in Scandinavia, where sparse human population and high air density combined with terrain-induced wind flows make for an excellent region for wind farm development. According to IEA [1], 69 GW of wind energy were located in cold climates in 2012, with 10 GW/Year being built from 2013–2017.

A challenging aspect of operating wind farms in cold climate regions is that of ice accretion on the wind turbine blades. The accretion of ice affects the aerodynamic efficiency of the turbine, yielding lower power output. Additionally, increased loads and vibration can decrease the life of

turbine components [2] and ice throw from wind turbine blades can be hazardous to plant personnel. Therefore, the forecast of ice accretion is crucial for plant operators and dispatchers who must plan for wind operations at energy utilities and independent power producers. As a result, ice prediction methods have been developed in the form of short term forecast models [3] and regional ice climate atlases [4]. Although much work has been done in the past decade to develop ice prediction products, this forecast problem continues to be a complex challenge [5]. Icing models, such as that described in Makkonen [6], have been developed and tested in lab settings, but they rely upon prognostic meteorological data input from numerical weather prediction (NWP) models in order to produce predictions of ice load. The ice load forecasts are sensitive to changes in low level liquid water content (LWC), wind magnitude, temperature, and humidity [6].

On-shore wind farms in Scandinavia are often located within regions of complex terrain and near heterogeneous patches of boreal forest. In these regions, the typical icing event occurs due to low level clouds containing supercooled liquid water intersecting terrain, impacting the turbine structure [3]. Molinder et al. [5] discusses the impact of errors due to localization that can occur in regions with subgrid complex topography and mitigating these errors with an ensemble method. Land cover representation and the modeling of surface-atmosphere fluxes in areas of heterogeneous land surface cover has been a hotly contested topic for decades, with papers focusing on the sensitivity of the forecast of temperature and wind to changes in surface albedo [7] and roughness [8]. Several studies have focused on the impacts of land surface type on the timing and evolution of low level clouds and fog [8–10].

The aim of this study is to test the sensitivity of the NWP model forecast of wind turbine icing events to changes in subgrid representation of vegetation fraction. A single-column model is used in this analysis, which uses the same physics and parameterization packages as its 3D NWP cousin, but it integrates in time only vertical fluxes in a vertical column covering one horizontal grid box. While they do not allow the horizontal advection of upstream large scale meteorological variables into and out of the atmospheric column, single column models allow the user to change certain model parameters while holding large scale conditions constant. This has several advantages: first, as subgrid conditions are only integrated in one column, the computational expense is cheap relative to 3D models, allowing for one to run several experiments; second, as large scale conditions are held constant, analysis of subgrid responses to changes in parameters is less unwieldy. As a result, we were able to run several simulations with the same initial conditions, only changing the vegetation cover in each. The output from the experiments was then used as input into an icing model. From this, we could analyze the sensitivity of the turbine icing forecast to vegetation cover. In addition, we were able to test the impact of the icing forecast to vegetation fraction when two different surface boundary layer (SBL) parameterization schemes were used: a single-layer scheme and a multilayer scheme with explicit levels within the forest canopy.

Section 2 of this paper will describe the models used as well as the experimental design. Section 3 will present the results of the experiments in terms of both the icing model and its input variables, as well as show the response in two specific cases. Section 4 will discuss the results and Section 5 will conclude.

## 2. Method

The model used in this study—Modèle Unifié Simple Colonne (MUSC) [11]—is a single column version of the HARMONIE-AROME (cy40h.1.1.1, HIRLAM-consortium, International) configuration of the ALADIN-HIRLAM 3D NWP system [12] developed by Météo France as a testbed for NWP developers. HARMONIE-AROME was developed from the AROME mesocale model, itself born as part of the ALADIN (Aire Limitée Adaptation Dynamique Développement International)-HIRLAM (HIgh Resolution Limited Area Model) consortium. ALADIN-HIRLAM is a collaboration of 26 countries in Europe and North Africa with the aim to improve short-range weather prediction in this region. HARMONIE-AROME itself is a convection-permitting non-hydrostatic limited area model used

in operational forecast offices in Sweden, Norway, Denmark, Ireland, The Netherlands, Finland, Spain, Iceland, Estonia, and Lithuania. HARMONIE-AROME uses the microphysics scheme based on the work described in Pinty and Jabouille [13] and Lascaux et al. [14]; the microphysics parameterization in HARMONIE-AROME is a one-moment bulk scheme that includes water, cloud ice, snow, and graupel as prognostic variables. This is combined with the condensation parameterization that is described in Section 2b of Müller et al. [15], which addresses a challenge with the microphysics scheme in which ice formed too quickly at temperatures above $-20\ °C$ [12] MUSC contains the same physics and parameterization packages as HARMONIE-AROME with the exception of horizontal advection being ignored in the simulation; although MUSC (cy40h1.1.1, HIRLAM-consortium, International) contains an added capability to implement large scale forcing based on a horizontal thermal gradient. The result is a computationally cheap research tool that can be used to understand vertical turbulent exchanges or test the performance of new parameterization schemes that may be incorporated in new versions of 3D NWP models [16]. MUSC was configured while using similar settings to the operational version of HARMONIE-AROME used at the Swedish Meteorological and Hydrological Institute (SMHI). This configuration uses a 2.5 km horizontal resolution with 65 vertical levels.

*2.1. Surface Model*

The surface model used in MUSC is version 7.3 of the Surface Externalisee (SURFEX) externalized surface scheme [17]. SURFEX (V7.3, Météo France, Toulouse, France) is a package of physical parameterizations that calculate the subgrid energy exchange between the surface and atmosphere. When SURFEX is coupled with an atmospheric model, forcing data from each gridpoint of the model atmosphere is sent to a series of models that calculate fluxes of momentum, sensible heat, latent heat, gas-, and aerosol feedbacks for each of four tiles that represent a different surface type: sea, lake, nature (consisting of vegetation and soil), and urban areas [17]. In addition to the fraction of surface type tiles, SURFEX has a capability that can further divide the nature tile into up to 12 so-called patches. An average of each individual turbulent flux is then calculated, such that it is weighted by the fraction of each surface type and the resulting flux is sent back to the atmosphere, along with information about surface roughness, albedo, emissivity, and surface temperature. In SURFEX version 7.3, the surface type fractions are determined from ECOCLIMAP II (Version II, Météo France, Toulouse, France) [18], which combines satellite and land maps to create a database of surface descriptions at 1-km resolution. The surface description from ECOCLIMAP II, upper air temperature, specific humidity, horizontal wind components, pressure, total precipitation, aerosol concentration, gas concentration, downwelling short- and longwave radiation from the atmospheric model are sent to SURFEX at each model time step.

Available within the SURFEX parameterization package is a one-dimensional (1D) multilayer surface boundary layer (SBL) scheme developed by Masson and Seity [19] (see schematic in Figure 1b). When the Masson and Seity [19] canopy scheme is not activated, SURFEX uses a single layer scheme coupled to the lowest atmospheric model level while using a variation of Monin–Obukov similarity to calculate wind, temperature, and humidity profiles in the SBL (see schematic in Figure 1a).

Masson and Seity [19] proposed that the single layer scheme, while accurate in flat, homogeneous landscapes, failed to accurately estimate mass and momentum fluxes within tree canopies, as the lowest atmospheric model level is assumed to be above the vegetation. Multilayer SBL schemes, with explicit atmospheric model levels within the tree canopies, provide the capability to calculate more accurate atmospheric profiles within the tree canopy, but are computationally expensive when used in operational forecast models. Masson and Seity [19] proposed a compromise multilayer SBL scheme that solves the atmospheric governing equations by utilizing the subgrid turbulence parameterization scheme combined with a large scale atmospheric term and canopy drag terms that were calculated from the leaf area index (LAI) of the forest. The turbulence scheme used in MUSC and HARMONIE-AROME is called HARMONIE with RACMO Turbulence (HARATU) [12]. HARATU (cy40h1.1.1, HIRLAM-consortium, International) computes a prognostic turbulence kinetic energy (TKE) equation on half-levels using separate diagnostic length scales for heat and momentum [12],

providing the basis for flux computations within SURFEX. The large scale forcing term is the tendency of wind, temperature, and humidity at the lowest atmospheric level. Below the lowest atmospheric level, the aforementioned variables and turbulent kinetic energy (TKE) are calculations taken from the turbulence parameterization scheme used in the NWP system. The aim of the Masson and Seity [19] multilayer SBL scheme is to provide a computationally cheap solution to improve forecast performance in forested areas. It should be noted that, in their analysis of the performance of HARMONIE-AROME (cy40h.1.1.1), Bengtsson et al. [12] states that operational experiences found that the Masson and Seity [19] canopy SBL scheme yields temperatures that are too low in stable conditions with weak winds, which is contrary to what was found in Masson and Seity [19].

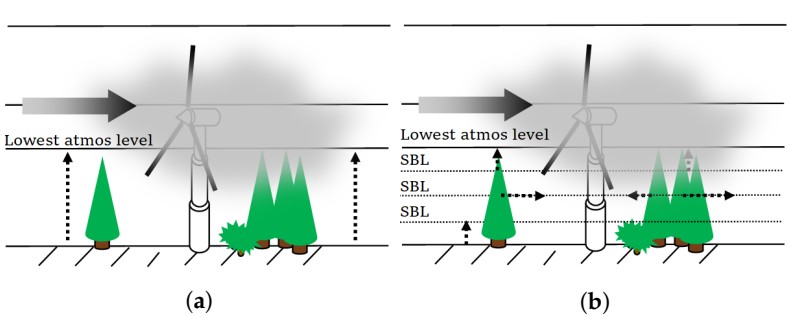

**Figure 1.** Examples of different surface boundary layer (SBL) schemes: (**a**) single layer SBL and (**b**) multilayer canopy scheme described in Masson and Seity [19].

## 2.2. Icing Model

The output from each simulation was run through an icing model based on the one described in Makkonen [6] and adapted to wind turbine use in Bergström et al. [20]. This is the same icing model used by Molinder et al. [5]. The model is based upon the equation:

$$\frac{\mathrm{d}M}{\mathrm{d}t} = \sum_{i=1}^{5} \alpha_1 \alpha_2 \alpha_3 wVD - L, \tag{1}$$

where $M$ is the mass of the accumulated ice in kg, $t$ is time, $w$ is the liquid water content, $V$ is the wind speed, $D$ is the diameter of the cylinder, and $\alpha_1$, $\alpha_2$, and $\alpha_3$ are the collision, sticking, and accretion efficiencies, respectively. $L$ is an ice loss term that combines a series of functions to take ice shedding, melting, sublimation, and wind erosion into account [5]. Equation (1) uses as input from the NWP output wind speed, temperature, pressure, cloud water, ice, rain, graupel, snow, and specific humidity. Collision efficiency $\alpha_1$–or the fraction of supercooled particles that make contact with the structure–is determined by the concentration of the particles, size distribution of the particles, wind magnitude, and size of the structure. The size distribution of the supercooled water droplets is estimated using the median volume diameter (MVD) calculation described in Finstad et al. [21]. As the specific MVD is unavailable as NWP output, it was estimated using the liquid water content output and assuming that the number concentration of liquid water particles is constant at 100 cm$^{-1}$ following Molinder et al. [5]. The sticking efficiency was determined from the work that was conducted by Kringlebotn Nygaard et al. [22], using $\alpha_2 = 1/V^{0.75}$. The accretion efficiency $\alpha_3$ is unity when water sticking to the surface of the blade freezes immediately, but is less than unity if the water takes time to freeze since some water is assumed to fall off the blade during that process. The icing model used in this study also includes functions that estimate ice loss due to sublimation, shedding, melting, and wind erosion. The mass of sublimation per unit time is calculated by solving Equation (19) in Mazin et al. [23], using relative humidity and wind speed from the NWP model. Shedding is estimated

using the same method as Molinder et al. [5], who multiplied a constant of eight with the melting term, which is itself calculated using the energy balance equation. The wind erosion is also calculated following Molinder et al. [5], who multipiled an hourly rate coefficient of 10 g m$^{-1}$ (m s$^{-1}$)$^{-1}$ for wind speeds greater than 5 m/s. The wind erosion is zero for wind speeds below 5 m/s. Note that the icing model is highly nonlinear, depending upon changes in the cross sectional size of the cylinder as the ice mass increases (impacting the Nusselt number, which is used to estimate the heat balance of the blade [23]), changes due to insulation effects from existing ice, wind speed, water particle size, and water content. In addition, Molinder et al. [24] describes the uncertainties in the formulations for ice shedding, MVD, wind erosion, $\alpha_2$, and $\alpha_3$. While these uncertainties should be considered, the details of these are beyond the scope of this study.

## 2.3. Experimental Setup

MUSC was initialized from a point in northern Sweden using 3D HARMONIE-AROME forecasts of six different low level in-cloud icing events in 2013 and 2014 in order to test the sensitivity of the model forecast to the vegetation fraction at the model's lower boundary. The aim here was to examine the dynamic and diabatic response of the model to changes of vegetation fraction under a range of differing meteorological conditions and incoming solar radiation profiles. This way, we could examine the response of the icing model to vegetation fraction changes when forced with the NWP output. From this information, one can analyze the icing model input variables to find the contribution of diabatic and dynamic components to the icing model response and the authors could look at the impact of vegetation fraction on the evolution of these events.

The initial conditions were chosen from relatively well-forecasted icing events described in the work done by Molinder et al. [5]. Additionally, the cases chosen represent a wide range of top of atmosphere (TOA) downwelling shortwave (SW) radiation profiles from both early and mid-season icing events in order to test the response due to changes in albedo. The cases chosen for the study are listed in Table 1 with a summary of the meteorological conditions of each case. Max TOA SW Down represents the maximum downwelling shortwave radiation at the top of the atmosphere for the duration of the event.

**Table 1.** Cases from which initial profiles were taken from the three-dimensional (3D) model.

| 3D Model Date | MUSC Init Hour | Event Type | Snow Cover | Max TOA SW Down | Initial Wind Speed at 117 m | Initial Temperature at 117 m |
|---|---|---|---|---|---|---|
| 24 December 2013 0600 | 0900 | Lifting fog | Yes | 132.5 W/m$^2$ | 13.7 m/s | 266 K |
| 5 January 2014 0600 | 0900 | Lifting fog | Yes | 157.1 W/m$^2$ | 4.7 m/s | 270 K |
| 5 January 2014 1800 | 2100 | Lifting fog | Yes | 157.1 W/m$^2$ | 5.7 m/s | 269 K |
| 18 October 2014 0600 | 0900 | Elevated cloud/Lifting fog | Yes | 449.6 W/m$^2$ | 1.2 m/s | 268 K |
| 22 October 2014 0600 | 0900 | Elevated cloud/Lifting fog | Yes | 417.9 W/m$^2$ | 9.9 m/s | 267 K |
| 23 October 2014 0600 | 0900 | Elevated cloud/Lifting fog | Yes | 410.0 W/m$^2$ | 13.9 m/s | 269 K |

For each case, the initial conditions remained the same, with the only adjusted variable being the vegetation percentage. For every case, MUSC was initialized from the +3 h output of the 3D model run, spun up for 3 h, and integrated for 48 h, with 60 s timesteps. The simulation was run for 48 h in order to analyze the consistency of any diabatic impacts over more than one diurnal cycle.

Table 2 lists the fractions of the different ground cover types for each experiment.

**Table 2.** List of ECOCLIMAP II cover fractions for each experiment.

| COVER 301 (Tundra) | COVER 302 (Broadleaf Forest) | COVER 310 (Boreal Forest) | BARE GROUND |
|:---:|:---:|:---:|:---:|
| 0% | 0% | 0% | 100% |
| 2% | 2% | 7% | 90% |
| 3% | 3% | 13% | 80% |
| 5% | 5% | 20% | 70% |
| 7% | 7% | 27% | 60% |
| 8% | 8% | 33% | 50% |
| 10% | 10% | 40% | 40% |
| 12% | 12% | 47% | 30% |
| 13% | 13% | 53% | 20% |
| 15% | 15% | 60% | 10% |
| 17% | 17% | 67% | 0% |

The surface cover from the raw 3D model output featured 86.3% total vegetation cover, mostly including broadleaf (22.7%) and boreal forest (54.7%) native to northern Sweden with a tree height of 14.08 m. The raw surface description at the location of interest was divided into three cover types, including Tundra (ECOCLIMAP II COVER 301), Broad leaf forest (COVER 302), and Boreal Forest (COVER 310). The Tundra surface type includes 27% coniferous forest, 44% grassland, 9% broadleaf forest, 8% park land, 6% rocks, and 6% bare soil. The broad leaf forest is described in ECOCLIMAP II as an area with 85% broadleaf forest, 7% coniferous forest, 3% park land, 3% grass, 1% rocks, and 1% bare soil. The largest contribution was from the boreal forest surface type, which includes: 73% coniferous forest, 10% broadleaf trees, 6% park land, 5% rocks, 5% bare soil, and 1% cropland. In each successive simulation, 10% of bare ground (ECOCLIMAP-II COVER 538) was included in the total surface cover fraction while the vegetation was decreased proportionally. The bare land representation in this study assumes a homogeneous patch of soil with a visible albedo of 0.28. This bare ground representation was an idealized choice for the experiment and it does not necessarily depict the soil conditions at the location the simulation was initialized from. Because snow was on the ground in all simulations and any low vegetation would be inactive in the subfreezing conditions, we were only concerned with the roughness elements associated with the forest cover. As such, we considered the bare soil in this study to be a sufficient representation for testing sensitivities. The resulting grid cell total albedo for each experiment as a function of vegetation cover is shown in Figure 2. The values that are shown in Figure 2 indicate the total contribution of vegetation, snow, and bare ground to the albedo of the grid cell. The bars indicate the range of albedo values for every time step of every simulation analyzed. Although the albedo varied somewhat from each case due to differences in snow age, each simulation had qualitatively the same decreasing response with increasing vegetation cover.

Each simulation was run with both the single and multilayer SBL schemes described in Section 2.1 in order to test the sensitivity of the experiments to the type of SBL scheme. In order to determine the subgrid roughness length $z_0$ to be coupled with the atmosphere, the SURFEX code weights each fraction of cover type $i$ within each grid cell, summing $n$ cover elements using the following:

$$a = \sum_{i=1}^{n} \frac{1}{\ln(z_j/z_{0_i})^2 \Gamma_i},$$ (2)

where $z_{0i}$ is the roughness length for each cover type, $\Gamma_i$ is the weight for each vegetation fraction, and $z_j$ is the averaging level. The average subgrid $z_0$ is then determined using

$$z_0 = z_j e^{-\frac{-a}{\Gamma_{tot}}},$$ (3)

where $\Gamma_{tot}$ is the sum of all surface cover weights. When the multilayer canopy scheme is used, $z_j$ is the height of the lowest canopy level. Otherwise, $z_j$ is the height of the lowest atmospheric level. More information about subgrid averaging of surface properties in SURFEX can be found

in Le Moigne et al. [17]. When the single layer SBL scheme is used, the roughness length in the SURFEX output represents $z_0$ aggregated over all surface cover types within the grid cell and it is an accurate representation of the surface effects felt by the model atmosphere at its lowest level. In contrast, because the multilayer SBL scheme uses canopy drag terms rather than roughness length to determine frictional effects related to forests, $z_0$ is only aggregated for bare ground and low vegetation when this SBL scheme is used. Therefore, the SURFEX output for grid cell average $z_0$ when using the multilayer SBL scheme appears as a low value relative to that of the single layer scheme. As a result, an effective roughness $z_{0eff}$ was calculated for both the single layer and multilayer SBL scheme experiments while assuming a logarithmic wind profile [25] from the surface to the lowest model level,

$$z_{0eff} = \frac{12}{e^{k\overline{u(12)}/u_*}},$$ (4)

in which $k = 0.4$ is the von Karman constant, $u_*$ is the friction velocity, and $\overline{u(12)}$ is the wind speed at the lowest model level of 12 m averaged for all cases for each respective SBL scheme. As the friction velocity was not explicitly included in the model output, this was calculated from the surface momentum flux using the relationship defined in Stull [25]:

$$u_* = \sqrt{\overline{u'w'}_s},$$ (5)

where $\overline{u'w'}_s$ is the surface momentum flux calculated in SURFEX and averaged over all cases for each respective SBL scheme.

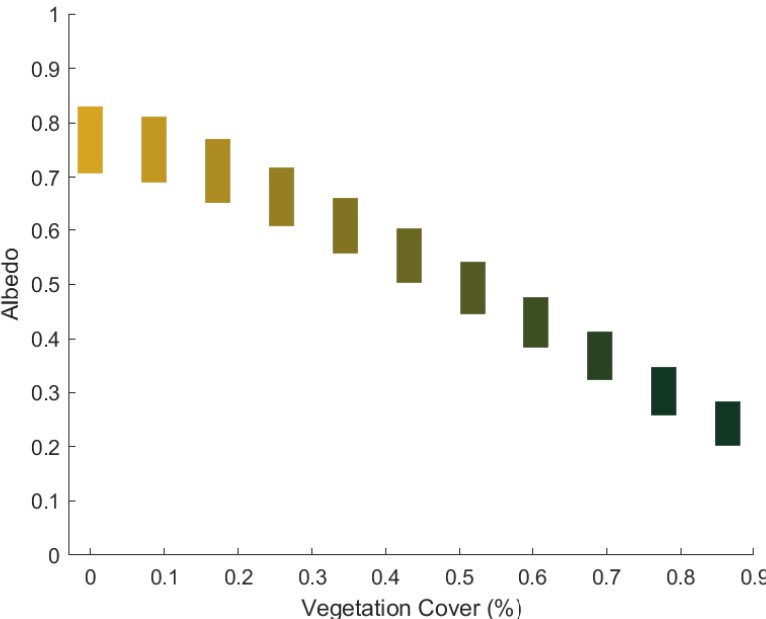

**Figure 2.** Total albedo as a function of vegetation percentage for each experiment. Bars denote the range of values for the albedo.

Figure 3 shows $z_{0eff}$ as a function of vegetation fraction when using the single layer and multilayer SBL scheme, respectively.

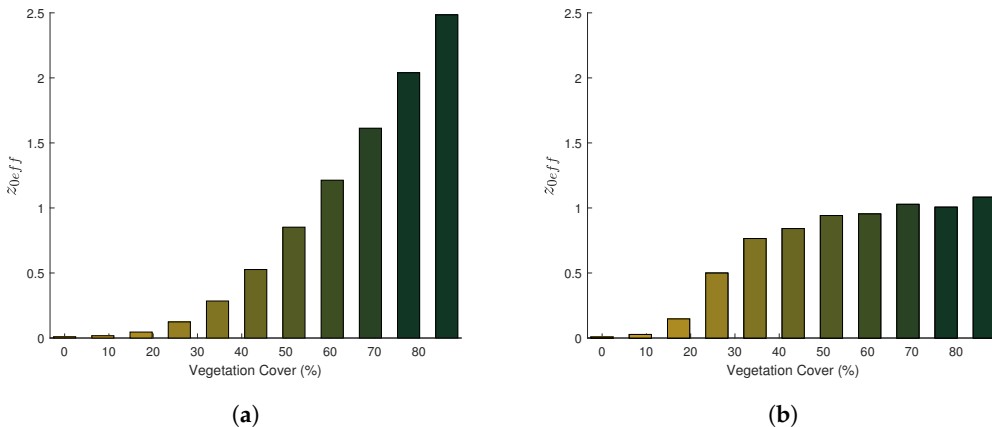

**Figure 3.** Effective roughness length $z_{0eff}$ in meters for single-layer (**a**) and multilayer SBL scheme (**b**) as a function of vegetation percentage for each experiment. The colors represent increasing vegetation fraction.

Note the difference between $z_{0eff}$ as a function of vegetation between the two SBL schemes. While the formulation for surface roughness described in Equation (4) assumes a logarithmic profile—which is not always the case—it is assumed to be a reasonable average approximation for depicting the relative difference between the two SBL schemes. That said, the effective roughness length for the single layer SBL scheme increases steadily with increasing vegetation fraction. This is in contrast to $z_{0eff}$ for the multilayer SBL scheme, which sharply increases from 10 to 40% vegetation fraction and only slightly increasing with increasing vegetation fraction above that. The multilayer canopy approach—which includes a forest drag formulation—will lead to blockage in the lower canopy as the density of the forest increases. This leads to a decrease in effective roughness length and increase in displacement height, as shown in Jackson [26]. In Figure 3b, the effective roughness, without the use of displacement height is shown, and this is the reason for the levelling off of roughness lengths for vegetation fractions that are greater than 50%. The impact of the SBL formulation on the icing forecast will be examined in the next sections.

## 3. Results

### 3.1. Impacts on Icing Forecast

All of the simulations for each of the vegetation fractions were run through the icing model with both the single layer and multilayer SBL schemes used. The absolute values of these results are shown in Figure 4, in which the maximum ice mass is shown for the 48 hour period in which MUSC was run. The values relative to the reference case with 86.3% vegetation fraction are shown in Figure 5. With the default single layer SBL scheme (Figure 5a), the average icing mass response generally decreases as a function of increased vegetation fraction. This relative decrease is nearly linear, with a change of approximately −20% per 0.20 increase in vegetation fraction.

The winter cases (red, black, and blue markers) feature a generally decreasing trend in ice mass as vegetation increases; following the mean trend. The 24 December 2013 case (red markers) is an example of this linear ice accretion response and it will be further examined in Section 3.3.2. The October cases were a bit more variable, with strong nonlinearity that was connected to the evolution of cloud cover and associated variations in atmospheric stability. Of note is the 18 October 2014 case (cyan markers), which experienced a local maximum in icing mass near 0.60 vegetation fraction and a larger relative impact overall as vegetation fraction is increased. With the single layer SBL scheme, the total ice mass in the case initialized from 18 October 2014 was nearly 375% higher in the bare ground case as compared with the reference case with 0.86 vegetation fraction. While it is noted that the 18 October 2014 case experienced comparatively less icing than the other events, the impact of

surface cover on the evolution of cloud cover is noteworthy. As such, the 18 October 2014 case will be further examined in Section 3.3.1.

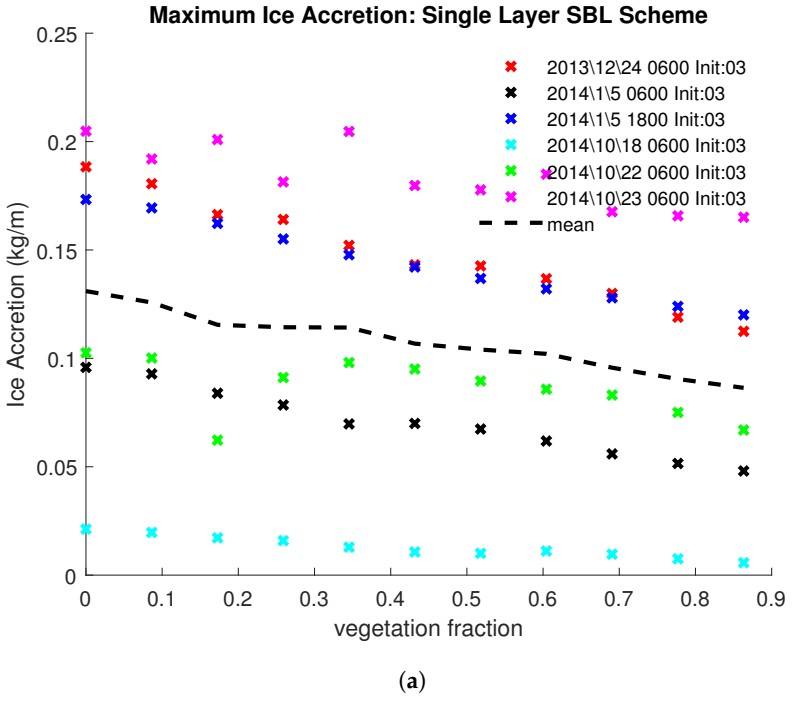

(**a**)

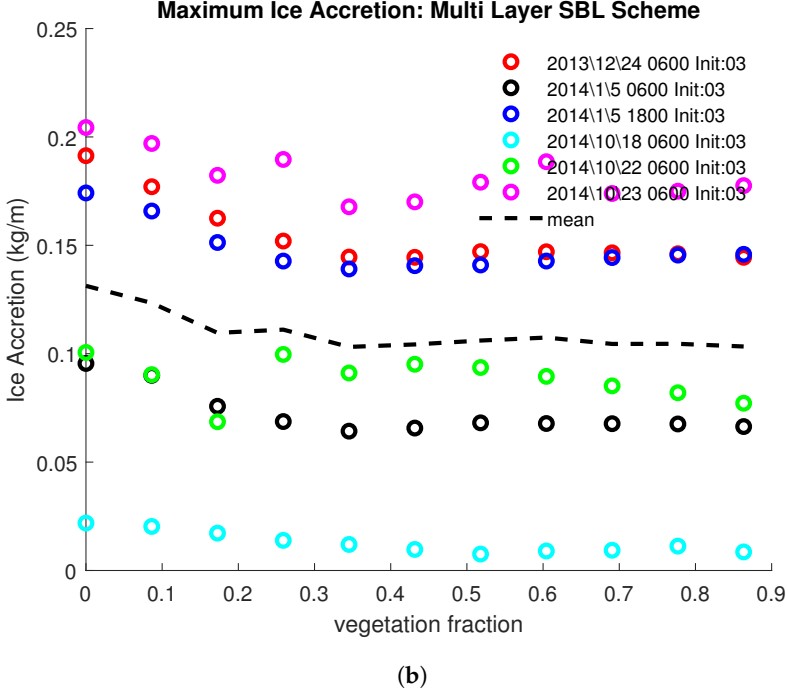

(**b**)

**Figure 4.** Absolute values of maximum ice accretion (kg/m) for the single- (**a**) and multi layer (**b**) SBL schemes. The black dashed line indicates the mean value for each vegetation fraction.

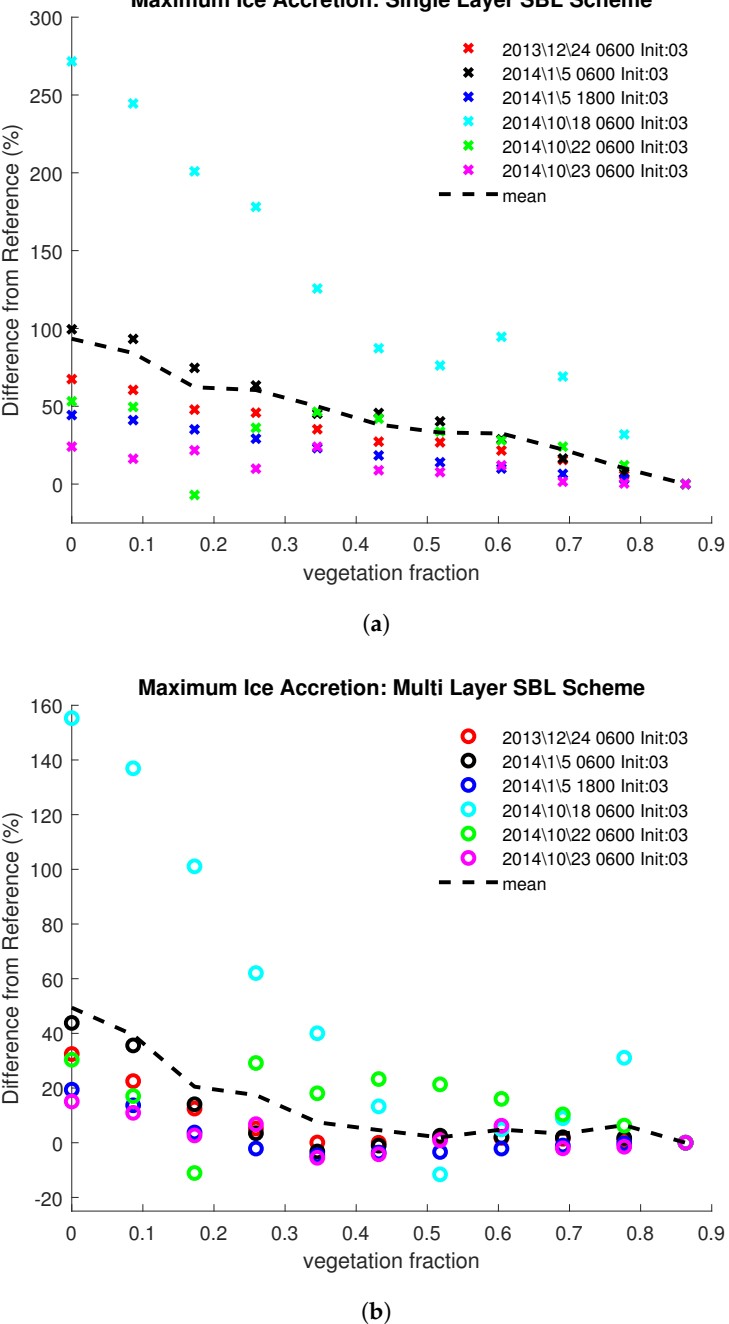

**Figure 5.** Percent difference of maximum ice mass accretion during the simulation from the 86.3% vegetation case as a function of vegetation fraction for all cases. (**a**) is the response with the single layer SBL scheme and (**b**) is the response with the SBL scheme turned on. The black dashed lined indicates the mean for each vegetation fraction.

With the multilayer SBL scheme activated (Figure 5b), the percent difference of maximum ice accretion compared with the 86.3% reference case was much lower than what was seen when the single layer SBL scheme was activated. As with the simulations using the single layer SBL scheme, the midwinter cases display a systematic decrease in ice mass accretion as the vegetation fraction increases. However, the decrease in ice accretion is limited to vegetation fractions less than 0.5. The ice accretion response to vegetation fraction is more variable in the October cases, with the bare ground case experiencing between 15 and 160% more ice accretion than the reference case. However, as vegetation fraction is increased above 0.40, the average ice mass impact remains between 0 and 20%

less than bare ground. The 24 December 2013 case follows the average trend quite well, with a nearly flat response in ice mass for vegetation fractions above 0.40. As with the experiments with the single layer SBL scheme, an outlier could be seen in the 18 October 2014 simulation, which experienced a near-linear decreasing trend in ice mass as vegetation fraction increased to 0.50, with a well-defined local maximum around 0.80 vegetation fraction. Both the 24 December 2013 and 18 October 2014 cases will be examined in further detail in Section 3.3.

### 3.2. Assessment of Icing Model Input Variables

Recall that the icing model described in Section 2.2 uses input from a NWP model to describe the flux of all phases of subfreezing water through the wind turbine sweep. As the efficiency of the accretion of ice on the wind turbine is dependent upon the content and velocity of the water molecules—as well as air temperature—this section will show the impact of vegetation on the input variables from the MUSC simulations. Here, we show the impact of vegetation fraction relative to the reference case for the NWP wind speed, temperature, and liquid water content at 117 m. The impacts of vegetation cover on rain, snow, ice, and graupel were not included here, as the values of these were negligible and very similar for all vegetation percentages.

In the experiments using the single layer SBL scheme, the mean wind magnitude impact follows the trend in roughness length (see Figure 3a) as vegetation is increased (Figure 6a).

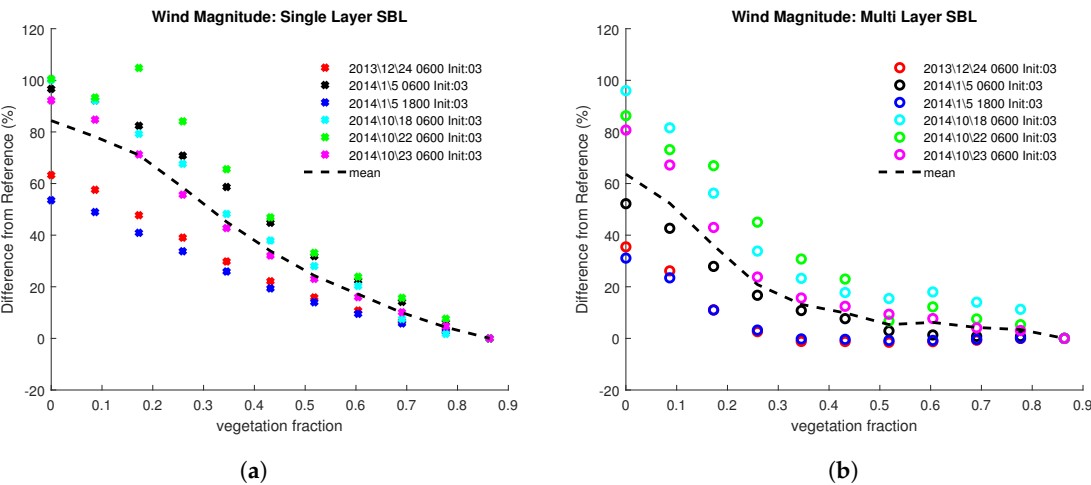

**Figure 6.** Percent difference of temporally averaged 117 m wind magnitude from the 86.3% vegetation case as a function of vegetation percentage for single layer SBL (**a**) and multilayer SBL scheme (**b**). The black dashed lined indicates the mean for each vegetation fraction.

The average wind magnitude response is nearly 85% greater for bare ground relative to the reference case. The general trend for wind speed is decreasing from bare ground to the reference case for individual cases as well, with the exception of the 22 October 2014 case (green stars). In the 22 October 2014 case, the wind magnitude is greater for the 0.19 vegetation cover case than bare ground. The average impact of vegetation percentage on wind magnitude in the simulations with the multilayer SBL scheme activated (Figure 6b) follows a similar pattern seen in the icing model trends for those simulations, with a nearly 65% decrease in wind speed response relative to the reference case up to 0.50 vegetation fraction and a flatter response at vegetation fractions higher than this. The mid winter cases (red, black, and blue circles) feature a nearly flat wind magnitude response with vegetation fractions above 0.50, but the slope of the October cases (magenta, cyan, and green circles) is steeper.

The average impact of vegetation fraction on temperature (Figure 7) is similar for both SBL schemes, with temperatures roughly 1 to 1.5 K cooler for the bare ground case relative to the highly vegetated reference. However, the spread amongst the cases examined is large. The winter cases experienced a very small temperature impact, owing to low levels of incoming solar radiation.

The temperature impact for the October cases as vegetation is increased varies from 1.25 K in the 23 October 2014 case to 3.5 to 4 K in the 22 October 2014 case.

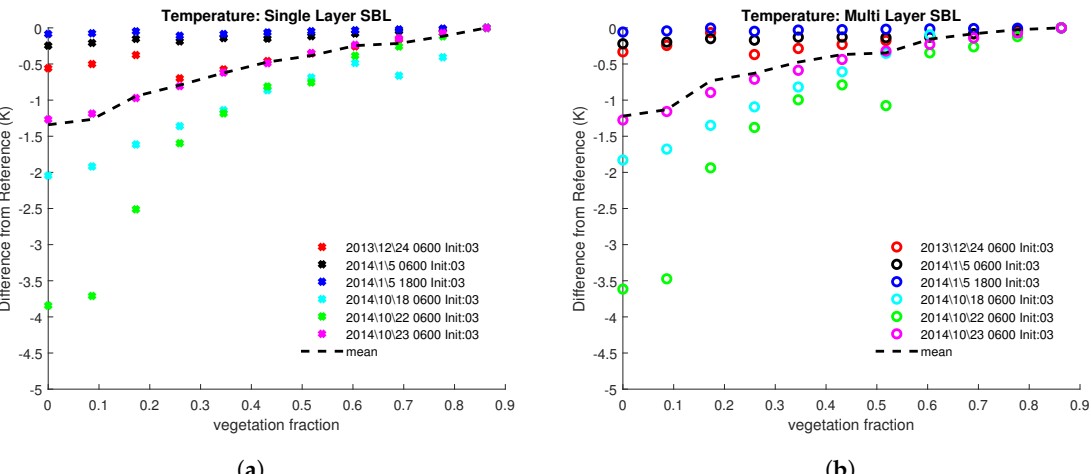

**Figure 7.** Percent difference of temporally averaged 117 m temperature from the 86.3% vegetation case as a function of vegetation percentage for single layer SBL (**a**) and multilayer SBL scheme (**b**). The black dashed lined indicates the mean for each vegetation fraction.

In both the single layer and multilayer SBL experiments, cloud water response (Figure 8) is not as systematic as the other variables examined, with the average response less than ±50%. The cloud water response as compared to the reference case in the winter cases is variable around 0%, with the case initialized on 24 December 2013 (red markers) showing a slight systematic 25% decrease between vegetation fractions of 0.3 and 0.9. The October cases experience highly variable responses to cloud water as vegetation fraction is increased, with a significant outlier of cloud water at turbine level for vegetation fractions between 0 and 0.20 in the 22 October 2014 case and less cloud water relative to reference for vegetation fractions up to 0.30 for the 23 October 2014 case. The cloud evolution in the October cases were strongly influenced by the diurnal forcing with increased solar radiation, enhanced by the impact of snow cover on the surface albedo in unvegetated parts of the grid cell. The resulting feedbacks due to cloud cover further increased the complexity of these processes. For instance, in the reference experiment for the 22 October 2014 case, the albedo effects due to the vegetation combined with a conditionally unstable atmospheric profile to form a convective cloud early in the simulation. With lower vegetation percentage, the relatively cool near-surface temperatures precluded day time cloud formation, but radiation fog did develop soon after 00:00 UTC. An opposite effect occurred in the 18 October 2014 case, which will be examined in the next section.

### 3.3. Cases

Several cases were analyzed in order to test the sensitivity of simulated events featuring varying diabatic and large scale influences to the represented forest fraction. Two cases will be examined in the next sections to provide further insight into the physical processes of the icing events and the modelling thereof. The case initialized from the morning of 18 October 2014 featured relatively weak synoptic dynamics under surface high pressure. The 24 December 2013 case featured stronger wind magnitudes owing to a more dynamic synoptic environment. The temporal evolution of meteorological variables for these cases will be examined to highlight the diurnal cycle under different surface albedos and roughness values. The diurnal response to the two different SBL schemes will be highlighted. While the cloud evolution is relatively similar with different SBL schemes, some important subtleties will be discussed.

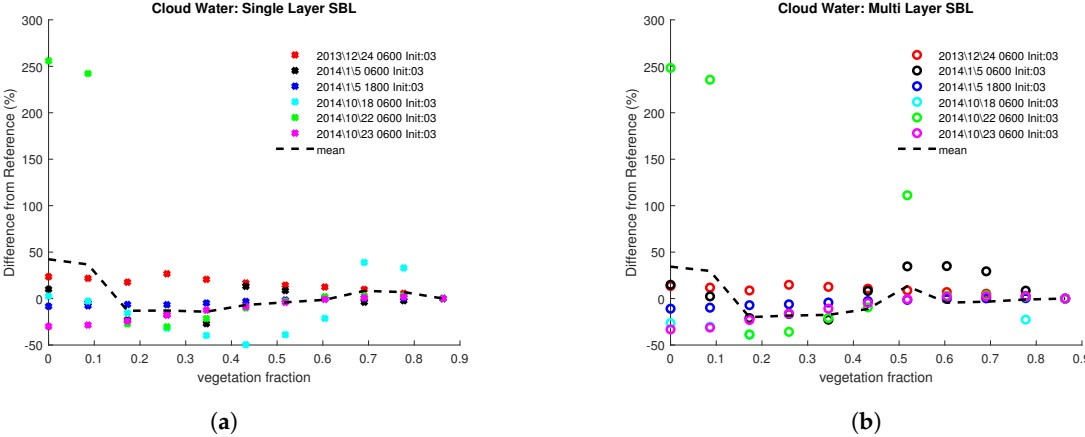

**Figure 8.** Percent difference of temporally averaged 117 m cloud water from the 86.3% vegetation case as a function of vegetation percentage for single layer SBL (**a**) and multilayer SBL scheme (**b**). The black dashed lined indicates the mean for each vegetation fraction.

### 3.3.1. 18 October 2014 Case

On the morning of 18 October 2014, a deep surface low pressure system was present in the north Atlantic with an occluded front stretching from southern Sweden into central Europe, where a weak low pressure system was present. In contrast, a region of high pressure was present over the northern Baltic Sea, with weak pressure gradients owing to light winds at the target area in northern Sweden. Figure 9 shows the 900 mb cloud fraction, wind field, and temperature at 09:00 UTC from ECWMF ERA5 reanalysis data. Surface temperatures were below freezing across the entire region on that day with a subsidence inversion present at 400 m. Winds were westerly across much of Sweden, with stronger south-southwesterly winds converging near the northern parts of the border with Norway. Low clouds developed over the next 24 h along the line of convergence, signalling the beginning of a near surface in-cloud icing event at the point of interest symbolized by a black dot on the aforementioned chart. It was at this point of interest that MUSC was initialized.

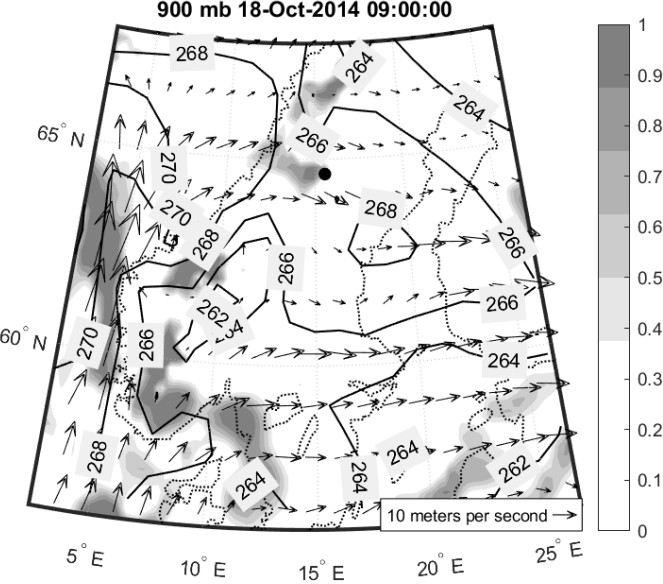

**Figure 9.** ECMWF ERA5 reanalysis: Cloud fraction (shading), wind vectors (m/s), and temperature (K; contours) at 900 millibars at 09:00 UTC 18 October 2014 over Scandinavia. MUSC was initialized at the location of the black dot.

The cloud water evolution for the 18 October 2014 case is greatly impacted by changing the vegetation cover in the single-column experiments. As vegetation cover is increased, the activation of turbine level cloud water is delayed into the early morning hours, with higher values occurring close to the ground. This is highlighted in Figure 10, which shows time height cross sections of cloud water for four vegetation fraction experiments that were initialized on 18 October 2014 with the single layer SBL (multilayer SBL simulation not shown).

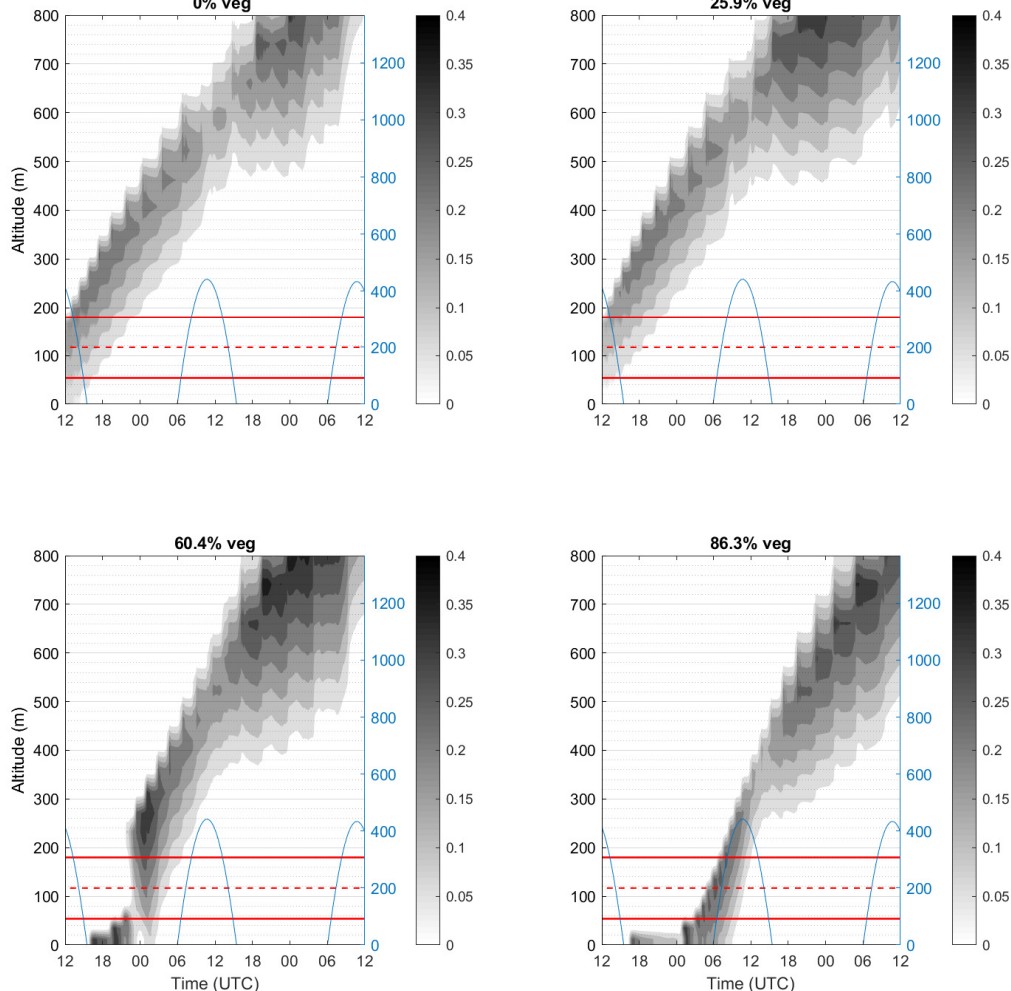

**Figure 10.** Time-height cross section of cloud water content (g/kg, filled contours) for the 18 October 2014 case for vegetation fractions of 0%, 25.9%, 60.4%, and 86.3%, respectively, using the Single Layer SBL scheme. Since the qualitative changes to the cloud evolution for each successive vegetation fraction in this case were roughly continuous, four vegetation fractions from bare ground to the reference case with the largest fraction of vegetation were shown here for illustrative purposes. Top of the atmosphere downwelling shortwave radiation (W/m², right *y*-axis) has been added to indicate the diurnal cycle. The red lines indicate the elevation of a hypothetical wind turbine with a hub height of 117 m (dashed lines) and a rotor diameter of 126 m (solid lines indicating turbine sweep).

One can see slight differences in icing intensity (Figure 11) between the two SBL schemes, with more subtle impacts to duration as the cloud lifts through turbine hub level. In the single layer SBL case (Figure 11a), other than an outlier in the 60.4% vegetation case in which icing intensity is

higher than the adjacent vegetation fractions between 00:00 and 03:00 UTC on 19 October 2014, the icing intensity and duration generally decreases with increasing vegetation fraction. Using the multi layer SBL scheme (Figure 11b), the general icing behavior is similar as in the single layer simulations; however, the icing intensity and duration increases slightly.

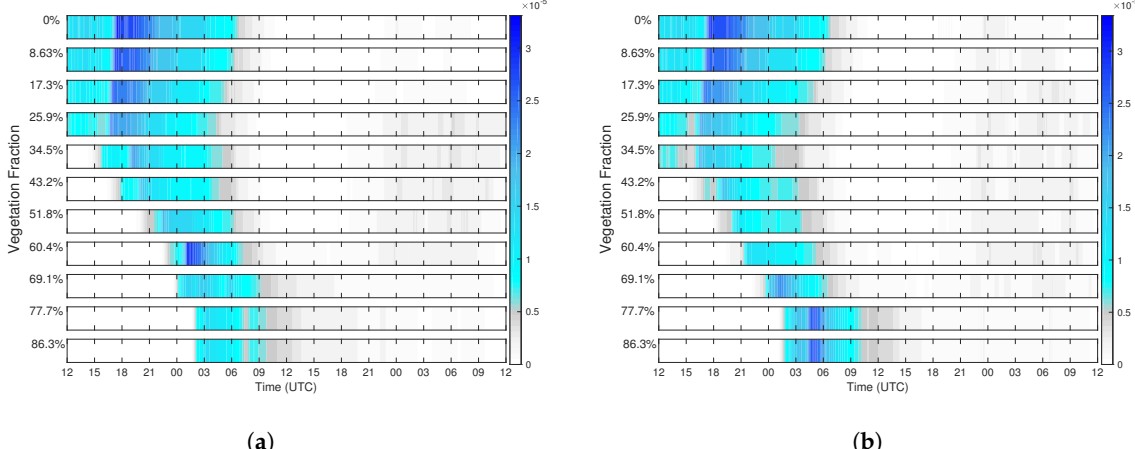

(**a**)                                                  (**b**)

**Figure 11.** Time steps for the single- (**a**) and multilayer (**b**) SBL schemes in which active icing occurred during the 18 October 2014 case for each vegetation cover fraction. Colors indicate the icing intensity (kg/m/minute).

The cloud deck develops soon after the single column model is initialized for the experiments with lower vegetation cover percentages and persists through the daytime hours, but it is not present in the higher vegetation cases until later in the simulation. This is likely due to a higher sensible heat flux owing to a greater absorption of downwelling radiation with the lower albedo of higher forest fractions and associated dry convective mixing under a capping inversion precluding low cloud development early in the simulation. The temperature at turbine level reflects this (Figure 12a), as temperatures in the experiments with higher vegetation fractions peaked at up to 4 K warmer than the bare ground case on 15:00 UTC on 18 October 2014. With the single layer SBL scheme (dashed lines), low level temperatures drop off dramatically after 00:00 UTC for vegetation fractions higher than 60.4%. The cloud water (Figure 12b) increases in these experiments as the air cools to condensation and fog lifts through the turbine hub elevation. The multilayer SBL (solid lines) developed slightly more cloud water than the single layer SBL between 04:00 and 07:00 UTC for the 77.7 and 86.3% vegetation fractions, which—combined with higher wind speeds at hub height—may explain the increase in icing intensity for those experiments.

Wind speeds were initially similar amongst all experiments Figure 13, but deviated after 18:00 UTC on 18 October 2014, with higher vegetation fractions predictably experiencing wind speeds generally up to 1 m/s lower than the bare ground case. An outlier can be seen in the single layer SBL 86.3% reference case, in which the wind speeds drop nearly to 0 m/s after 06:00 UTC on 19 October 2014. These low wind speeds occurring under stable conditions with the higher effective surface roughness associated with the single layer scheme presumably contributed to the lower icing intensity during those times relative to the multilayer SBL examples.

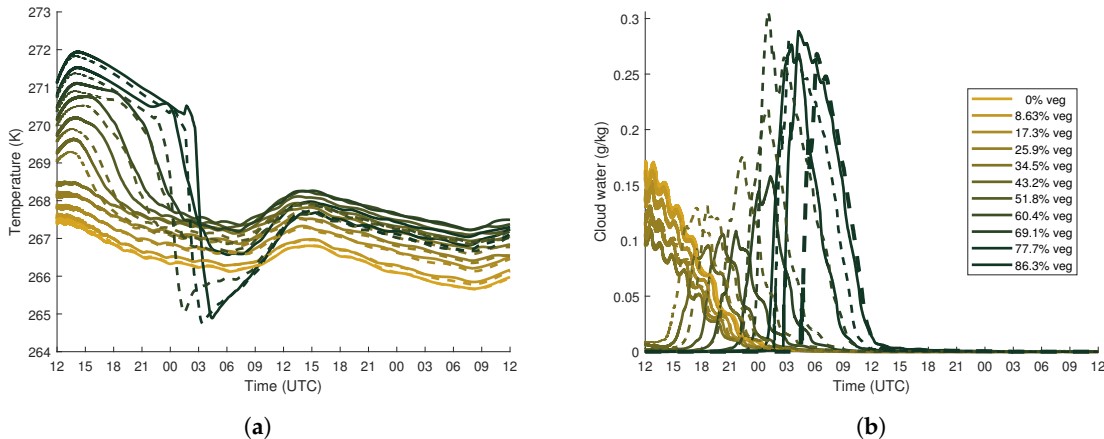

**Figure 12.** Time evolution of air temperature (**a**) and cloud water (**b**) for the 18 October 2014 case at 117 m for each vegetation percentage. Solid lines denote the experiments using the multilayer SBL scheme and the dashed lines denote the experiments using the single layer scheme.

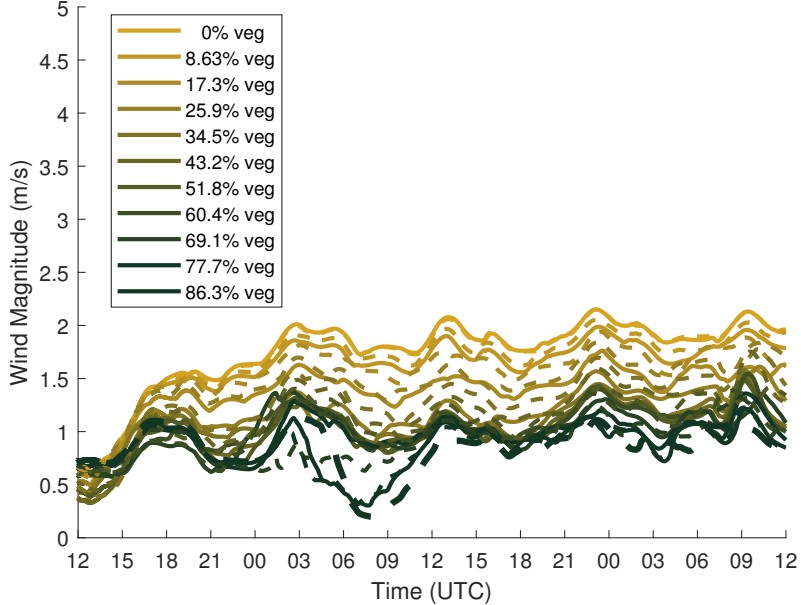

**Figure 13.** Time evolution of wind speed for the 18 October 2014 case at 117 m for each vegetation percentage. Solid lines denote the experiments using the multilayer SBL scheme and the dashed lines denote the experiments using the single layer scheme.

In all cases, these low level clouds lift through the turbine sweep and into the upper boundary layer throughout the simulation. In the cases with higher vegetation fraction, the clouds lift dramatically through the morning hours on 19 October 2014 as low level temperatures rebound. It is noted that, during the time that the supercooled clouds lift through the turbine sweep, the wind speeds increase and remain nearly constant through the rest of the simulation. While winds were relatively light in this case, this result is significant as icing is impacted by wind speed, as seen in the analysis in the previous sections. The evolution and magnitude of the liquid water content at turbine level (Figure 12b) varies considerably as a function of vegetation fraction and SBL scheme type, with a maximum of 0.3 g/kg for the single layer SBL simulation while using 69.1% vegetation fraction as the clouds lift through turbine level. Recall that the icing response in the 18 October case relative to the reference case was nearly flat between vegetation fractions of 0.50 and 0.70.

3.3.2. 24 December 2013 Case

On the morning 24 December 2013, a surface low pressure system was present off the coast of Scotland, with an associated occlusion and warm front stretching eastward into southern Norway and Sweden. An area of surface low pressure was present in far northern Sweden near the Arctic Circle. Figure 14 shows the regional wind field and cloud cover at 09:00 UTC from ECMWF ERA5 reanalysis data.

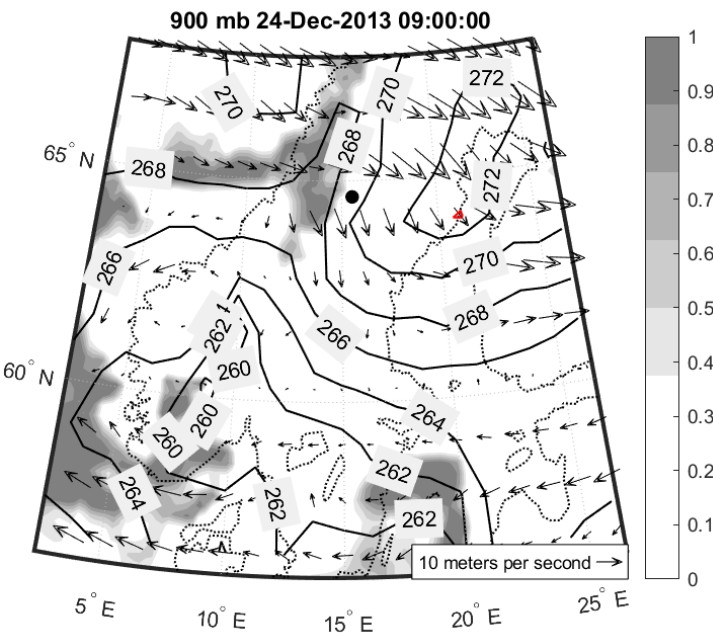

**Figure 14.** ECMWF ERA5 reanalysis: Cloud fraction (shading), wind vectors (m/s), and temperature (K; contours) at 900 millibars at 09:00 UTC 24 December 2013 over Scandinavia. MUSC was initialized at the location of the black dot.

Temperatures near a developing warm front over Baltic Sea were below freezing, but warming close to the freezing line toward the east. Relatively strong northwesterly winds that were associated with low pressure to the northeast were present in Northern Sweden, with a weak deformation zone near the Norwegian border. Near the line of convergence, a north-south oriented line of clouds formed. A low level icing event occurred at the point of interest over the next few days. Time-height sections of the cloud water content are seen in Figure 15.

In the single layer SBL reference experiment with 86.3% vegetation, a cloud forms at the surface slightly after 09:00 UTC on 25 December 2013 and lifts into the evening hours. As vegetation cover was decreased, the cloud evolution is similar, albeit progressively later into the morning. Clouds form in the 0% vegetation case at approximately 12:00 UTC on 25 December 2013. Figure 16 shows the ice duration and intensity for the 24 December 2013 case with the single and multilayer SBL schemes used. One can see the roughly 3 h difference of the onset of icing between the bare ground and 86.3% vegetation fraction reference case in the single layer SBL scheme (Figure 16a), which follows the cloud evolution. The icing duration in the initial event is also shorter for higher vegetation fractions. Using the multilayer SBL scheme (Figure 16b), the onset of icing accretion occurred after 12:00 UTC on 25 December 2013 and at roughly the same time for all experiments, but the duration was slightly shorter with increased vegetation fractions. Brief periods of snowfall occurred for all experiments after 00:00 UTC on 26 December 2013. The intensity of this snowfall and the timing of it was roughly the same for all experiments, but the impact on ice accretion was slightly different with increased vegetation fractions.

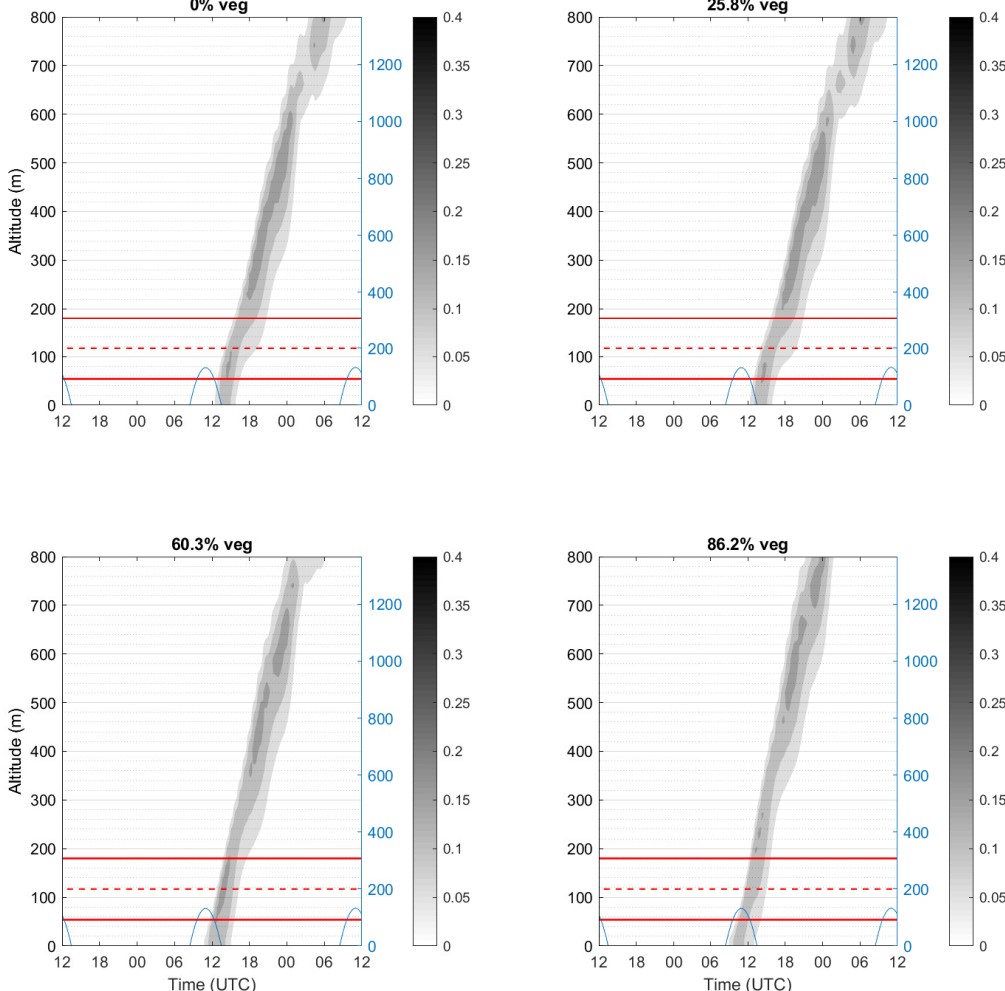

**Figure 15.** Time-height cross section of cloud water content (g/kg, filled contours) for the 24 December 2013 case for vegetation fractions of 0%, 25.9%, 60.4%, and 86.3%, respectively, using the Single Layer SBL scheme. Since the qualitative changes to the cloud evolution for each successive vegetation fraction in this case were roughly continuous, four vegetation fractions from bare ground to the reference case with the largest fraction of vegetation were shown here for illustrative purposes. Top of the atmosphere downwelling shortwave radiation (W/m$^2$, right y-axis) has been added to indicate the diurnal cycle. The red lines indicate the elevation of a hypothetical wind turbine with a hub height of 117 m (dashed lines) and a rotor diameter of 126 m (solid lines indicating turbine sweep).

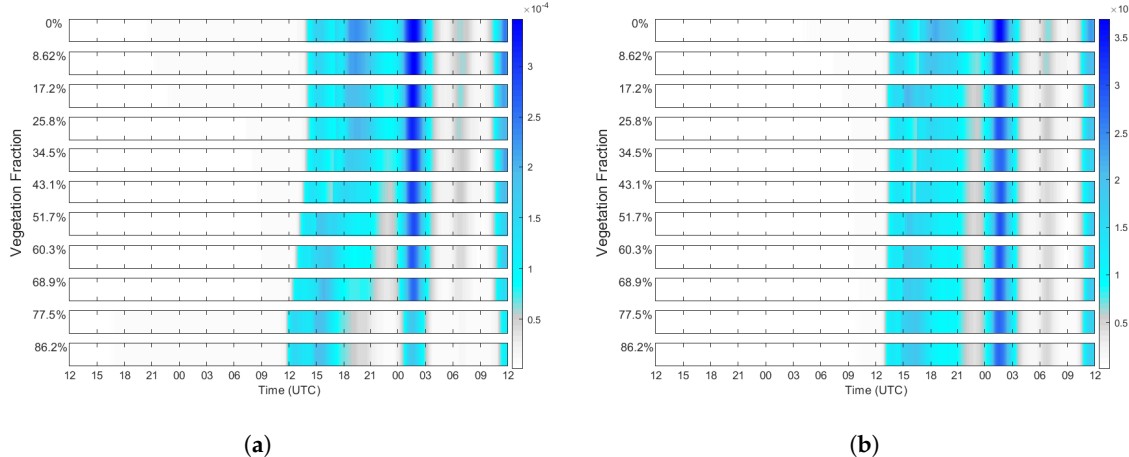

(**a**)
(**b**)

**Figure 16.** Time steps for the single- (**a**) and multilayer (**b**) SBL schemes in which active icing occurred during the 24 December 2013 case for each vegetation cover fraction. Colors indicate the icing intensity (kg/m/minute).

The wind speed at 117 m is seen in Figure 17 for each simulation and the SBL schemes. The initial wind speeds vary from nearly 12 m/s in the bare ground simulations to 6.5 m/s in the simulation with the highest vegetation fraction for the single layer SBL (dashed lines). The initial wind speed response with vegetation fraction predictably follows the effective surface roughness pattern for the multilayer SBL scheme (solid lines), with wind speeds decreasing between 11.5 m/s and 8 m/s as vegetation fraction increases to 34.5%. At vegetation fractions that are higher than 34.5%, the differences in wind speed for the multilayer SBL experiments are very subtle.

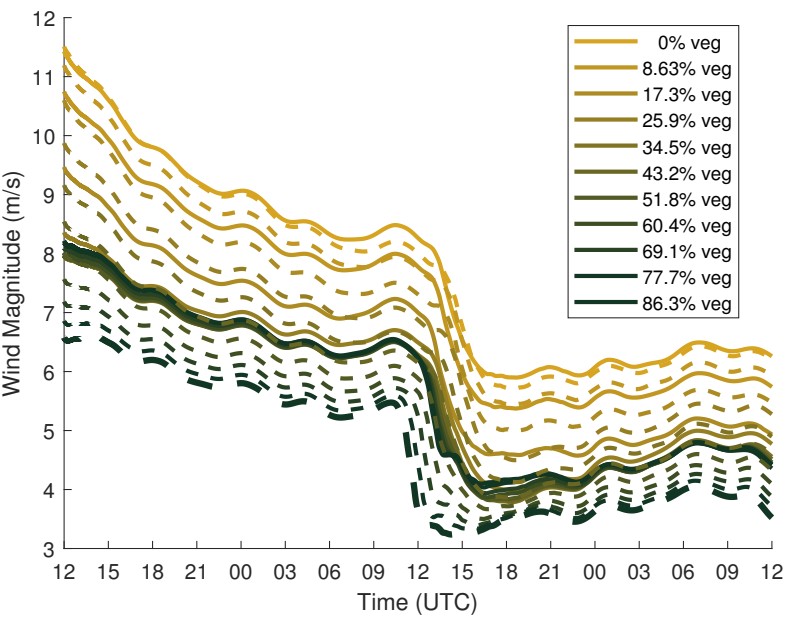

**Figure 17.** Time evolution of wind speed for the 24 December 2013 case at 117 m for each vegetation percentage.

The wind speeds for each simulation gradually decrease during the first 24 to 30 h before the boundary layer decouples, with wind speeds dropping steeply. This decoupling occurs earlier as a function of vegetation fraction for the single layer SBL scheme. The reference case with 86.3% vegetation experiences a drop in wind speeds roughly at the same time the cloud lifts through the turbine level. The multilayer SBL scheme experiments experience a similar drop in wind speed, albeit at roughly the same time for all vegetation fractions. The low level temperatures (Figure 18a)

cool to condensation as the boundary layer decouples and fog forms. The cloud water at 117 m peaks near 0.15 g/kg for all vegetation percentages. However, the timing of the onset of the supercooled water varies by about three hours for the single layer SBL case, with the onset occurring earlier in the higher vegetation cases (Figure 18b). The spike in cloud water at turbine hub height occurs at roughly the same time for all vegetation percentages when the multilayer SBL scheme is used.

Simulations with higher vegetation fractions relative to the bare ground case led to an earlier development of low level cloud cover. The clouds gradually lift during the day as the relatively lower albedo leads to an increase in upward sensible heat flux in response. In the bare ground cases, clouds formed later, lifting in a similar manner to the higher vegetation cases. Low incoming solar radiation levels led to weaker turbulent mixing than the 18 October 2014 case, allowing for fog to form in all cases, even during the daytime hours.

In the 24 December 2013 case, there is less downwelling shortwave radiation due to the low December sun angle at the latitude of the test location and a smaller difference in initial temperature between the experiments (Figure 18a). However, the 117 m wind speed (Figure 17) is higher in the 24 December 2014 case than in the 18 October 2014 simulation and the sensitivity to changes in vegetation is greater in this case. Recall that the icing response follows the trend in the relative decrease in wind speed associated with increased vegetation fraction and SBL representation. Additionally note that all experiments using the 24 December 2013 case showed roughly the same amount of cloud water as a function of vegetation percentage. Despite only subtle differences in maximum liquid water content, the ice mass still decreased in this case with the mean wind response.

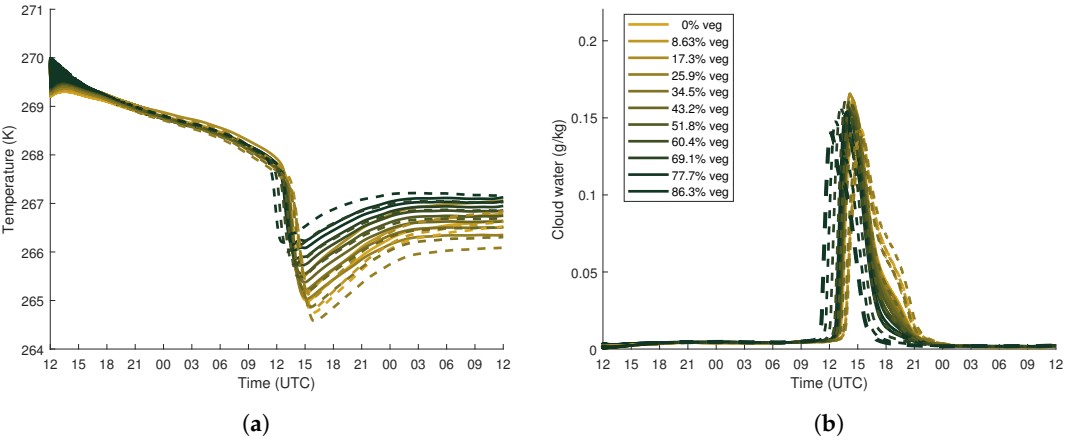

|           (a)           |           (b)           |

**Figure 18.** Time evolution of air temperature (**a**) and cloud water (**b**) for the 24 December 2013 case at 117 m for each vegetation percentage.

## 4. Discussion

The experiments show that vegetation representation has an impact on both the total accretion and evolution of ice formation during simulated wind farm icing events. The icing model is most sensitive to the impact of the effective roughness on wind magnitude at wind turbine hub height. The sensitivity of the icing model to wind speed is further highlighted when the SBL parameterization is changed to a scheme that aims to include tree canopy effects, in which the icing response also appears to have a strong correlation to surface roughness. With the Masson and Seity [19] multilayer SBL scheme used, the sensitivity of simulated ice accretion to vegetation cover seems to decrease at higher forest fractions. However, the sensitivity of simulated ice mass accumulation remains high at lower vegetation percentages. This suggests that the simulation of low level icing events may be impacted by not only the uncertainty in vegetation type, but by the surface model, which feature tuning parameters to accurately depict boundary layer processes [19]. When compared with the single-layer SBL scheme, the multilayer SBL scheme gives an effective roughness length response to changing vegetation cover that suggests increasing homogeneity with increased forest

canopy cover, which follows the response expected from theory (see Jackson [26]). In Section 2.1, Bengtsson et al. [12] was referenced, who discussed that forecasted low level temperatures were caused to be much too low in stable conditions when coupling the HARMONIE-AROME model to SURFEX with the multilayer scheme turned on. This suggests that, while the multilayer SBL scheme used in this study gives a surface roughness response that follows observations, proper tuning is required for the best results. Recent configurations of the HARMONIE-AROME model have moved away from the multilayer canopy scheme in favor of calculating surface fluxes over two separate patches of nature types within a grid cell before determining the weighted average flux over all other surface types. Brousseau et al. [27] suggested decreasing the height of the lowest atmospheric model level from 10 to 5 m in the related AROME model to explicitly resolve atmospheric layers within the tree canopy; they found that this precluded the need for the Masson and Seity [19] multilayer scheme. Diabatic effects due to surface albedo differences that impact the diurnal evolution of cloud cover also impacted the icing cases that were examined. This is evident in the 18 October 2014 case, which featured greater solar insolation than the other mid-winter cases. In this case, the impact of vegetation on the total ice mass was not as systematic as the mid-winter cases, with the cloud evolution impacted by the convective response to changes in surface albedo. This shows that, while the dynamic effects due to surface roughness remain dominant, the albedo effect has a highly relevant impact on the timing and evolution of the icing event. Recall that, in Section 2.3, the model surface is covered by snow, which enhanced the albedo of the bare ground contribution to each experiment. In a test running the 22 October 2014 case without snow cover (not shown), the impact of vegetation on the cloud evolution was reduced, suggesting a decreased heat flux response with vegetation changes. This is not surprising, as the impact of the representation of snow cover in the vicinity of forests on the surface energy balance has been examined for decades (see [7]).

As the icing model seems to be the most sensitive to changes in wind speed with increasing forest fraction, wind production can be impacted by the uncertainty of land cover representation and SBL parameterization in addition to the icing forecast. The ECOCLIMAP-II package [28] used in SURFEX is a 1-km resolution land surface database created from several remote sensing datasets of higher resolution. While land cover representation datasets of this resolution have been successfully used in global climate models and NWP models of relatively coarse resolution, some further consideration may be needed in the future as icing and wind production forecasts use NWP input from finer scale models. As one moves to higher resolution, one also must consider the heterogeneity, or patchiness, of the forest represented in the model forecast. The land cover that was represented in the experiments described in this study was assumed to be homogeneously distributed inside the grid cell. Several large eddy simulation (LES) studies (for example, [8,16,29–31]) have explored the varying impacts of land surface heterogeneties on boundary layer cloud processes. As forests surrounding onshore wind farms in Scandinavia often feature patchy heterogeneities due to nearby forestry operations and clearings for turbine equipment, and extension of this work is to examine how this patchiness of vegetation can impact wind turbine icing processes. With a properly-tuned SBL scheme, future mesoscale models may take advantage of the latest generations of surface description maps, such as the objective roughness approach (ORA) described in Floors et al. [32] that uses airborne lidar scans of tree height and density to create detailed snapshots of surface roughness. In addition to the local impacts of land use, nonlocal impacts to the icing conditions must also be examined. The single column model used in this study does not include the effects of horizontal advection, which can have downstream effects on the atmospheric profile. In an LES study of a wind field in neutral atmospheric conditions over a heterogeneous forest, Arnqvist et al. [33] showed that a large modeling domain is needed to accurately capture the changes in wind fetch due to upstream land surface heterogeneity. All of these considerations motivate our future work for using an LES model to capture nonlocal effects on icing conditions.

## 5. Conclusions

A single column model was used to test the sensitivity of ice accretion predictions at a high latitude location to subgrid land use representation from the meteorological input into the icing model. The single column model demonstrates that the icing on wind power turbines is sensitive to the forest fraction in a gridcell. In the mid-winter cases examined in which the downwelling solar radiation was relatively low, the icing forecast seems to be the most sensitive to the wind response due to surface roughness changes as forest fraction is increased.

In the early season icing case examined with increased diabatic forcing in the form of increased solar insolation, the trend of total predicted ice mass was less systematic than the mid-season cases, suggesting a sensitivity to the evolution of the in-cloud icing event. In this case, the cloud formation in the single column model was more transient and this impact can be seen in the ice mass predictions as vegetation increases. However, the general trend still appears to be sensitive to the wind response when surface roughness is increased.

One can see a similar response to wind magnitude in the icing prediction when simulating the cases using a more sophisticated SBL scheme in the single column model with a different calculation of surface roughness. While this suggests that uncertainty in the represented subgrid forest fraction can lead to errors in the wind turbine icing prediction, the surface layer model parameterization can also lead to uncertainty.

**Author Contributions:** E.J. performed the simulations, analyzed the data, and wrote this article; H.K. and E.J. developed the experimental design; H.K., J.A. and A.R. supervised E.J., suggesting revisions and providing their expertise when shaping this paper. All authors have read and agreed to the published version of the manuscript.

**Funding:** This research was funded as a part of Energimyndigheten contract number P44988-1 with support from STandUP for Wind.

**Acknowledgments:** The authors would like to thank our colleagues at SMHI and Uppsala University for support leading up to the completion of this manuscript, as well as Laura Rontu from the Finnish Meteorological Institute and Emily Gleeson of Met Éireann for support with the MUSC single column model. This work was partly conducted within STandUP for Wind, a part of the STandUP for Energy strategic research framework. We also thank the Swedish Energy Agency (Energimyndigheten) for funding this project.

**Conflicts of Interest:** The authors declare no conflict of interest.

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
