# Peer review of "Single Column Model Simulations of Icing Conditions in Northern Sweden: Sensitivity to Surface Model Land Use Representation"

_energies, doi:10.3390/en13164258_

Round 1
Reviewer 1 Report
Reviewer Comments to Energies-874625: Sensitivity of Mesoscale Model Forecasts of Icing Conditions to Land Use: Dynamic and Diabatic Impacts---Janzon et al
This manuscript uses the single column model based on a mesoscale model and an icing model to investigate the impacts of sub-grid vegetation fraction on the simulated meteorological field and on ice mass accretion at wind turbines. The authors conducted sensitivity tests using a set of different vegetation percentages for a total of six icing events in 2013 and 2014. Two surface boundary layers with a single layer and a multilayer canopy scheme were used in the model runs. The authors found that ice mass accretion of wind turbines is sensitive to the sub-grid vegetation fractions through wind and cloud water responses to vegetation changes. The topic is of interest considering the ice accretion impacts on wind energy and the need to understand the sub-grid land-atmosphere interactions in numerical weather models. The manuscript is well-structured, and the analysis is clear. However, there are some interesting results in the manuscript that not yet explained or discussed. I recommend publication after careful modifications are made. My comments are below:
Major Comments:
- Title: The title is a bit misleading. The authors used a single column version of the mesoscale model in their study. I am not sure it is fair to include ‘Mesoscale Model Forecasts’ in the title.
- Justification of the control experiment: The 0% vegetation case was defined as the control experiment. In my impression, the control experiment is always the experiment based on the ‘real’ data. In Ln 166, it is stated that the vegetation cover is 86.3% in the raw model outputs. Won’t it make more sense to use the 86.3% vegetation percent (somewhere between row 9 and row 10 in Table 2) as the control experiment? Another defect to use the unrealistic control experiment with 0% vegetation is that it makes the following comparisons difficult to interpret.
I see the authors used four different sets which include 86.3% for the two case studies later. This setup needs to be added in Section 2.3.
- Responses of percent differences compared with the control experiment in different cases: One interesting thing from the sensitivity tests is that the percent differences can be totally different under different events, especially for cloud water. I think it would be helpful to include some more explanations or discussions, for example after Ln 255. Will the different meteorological fields (e.g., humidity, temperature) in different cases impact their responses to vegetation variation?
- Reference style in the main text: The reference style in the main text needs some modifications. For example, in Ln 105, ‘scheme developed by [16]’ will be more intuitive if modified to ‘scheme developed by Masson et al. 2009 [16]’. Please read through the whole manuscript to make corresponding changes.
Minor Comments:
- Ln 14: ‘early season cases’---Do you mean ‘early winter cases’?
- Eq 1: The equation format is messed in Eq 1 above Ln 129.
- Figure 4: The field ‘maximum ice mass accretion’ should be included in the y axis title or the main title. I would suggest including in the y axis title, but it is up to the authors.
- Figure 5 and Figure 6: x and y axis titles are missing.
- Figure 9: It is not clear what the red lines stand for.
- Figure 10: y axis range can be reduced for better visualization of the curves.
- Figure 11: The right panel figure is cropped.
Reviewer 2 Report
This is a paper that investigates the effect of surface vegetation properties,
mainly roughness length and albedo on icing of turbines by running a single-
column model from mesoscale model output. It also includes a more complex
multi-layer canopy model in the comparison.
The findings are interesting and vary with meteorological conditions. The
methods and results are presented well and explained, and I think this paper
is worth publication. My comments are mostly related to clarifications and
some expansions I would like to see on what is written, so I regard this as
minor revision.
Minor Comments
1. line 80. Give a reference for the microphysics scheme used in the model.
2. line 130. alpha_3 (accretion efficiency) should be quantified for
completeness. Also does accretion depend on existing icing in some proportional
way?
3. line 161. Perhaps more info could be given on these cases (windy? cold
or near freezing?)
4. line 170. These albedos do not correspond to zero vegetation in Fig. 2
presumably due to snow cover? Does snow albedo only affect the unvegetated
part?
5. Table 2. What are Oural and Taiga? Unfamiliar words.
6. Figure 4. These show normalized values and the text has no indication of
absolute values. It would be good to know the degree of icing of these
cases relative to each other. Presumably the maximum occurs during the cloudy
periods in later figures. Does it disappear quickly after the cloud water goes?
7. Figure 5. It is clear that the multilayer scheme behavior goes flat for
higher vegetation amounts in the same way as wind and z0. Some text
explaining why this happens in that scheme would be useful.
8. Figure 11. Perhaps an indication could be added to one of these figures of
when the icing was occurring. Presumably the second day is irrelevant to icing
in this case?
9. Similarly to above point in Figure 15 for December case. However, Fig.15b
shows some cloud water at turbine level the whole time. Was there icing the
whole time?
Typos/Wording
1. Eq. (1) latex error
2. "solar insulation" several places in text should just be "insolation" which
is defined as solar.
Reviewer 3 Report
The manuscript entitled “Sensitivity of mesoscale model forecasts of icing conditions to land use: Dynamic and diabatic impacts” analyzes the impact of vegetation cover on the forecast of ice mass accretion on wind turbines. This kind of studies are relevant to the efficiency of wind energy, as well as the security of wind farm’s technicians.
The manuscript is well written, and I think that is in the scope of this journal. I highly recommend its publication after some minor changes.
General comments
When you give a general overview of all cases, the relationship of the variables with the vegetation cover is well described but sometimes you should include a better justification of the reasons for the observed variations.
On the other hand, you analyze two SBL schemes that show interesting results and differences but then when you analyze two particular cases you select only the simple SBL scheme and I think that you should include the multi-layer case, at least, in the discussion.
In the simulations you ignore horizontal advection and I think that you should discuss the possible impact of advection in the forecast of ice accretion.
Particular comments:
12: I think you refer to “insolation” and not “insulation”
Eq 1: There is a mistake in the equation “mathrm”
140: Please, provide a description for accretion efficiency “a3”
140: You should also provide a description of the ice loss term
141-145: This paragraph is redundant with the previous ones. Please rewrite it or consider if it is necessary.
Table 1: What is “max TOA SW down”? Is it a maximum value for the area or for the duration of the event?
170: Please specify that the model considers near infrared albedo
177: You indicate that there are differences in albedo from bare ground due to snow cover, but I am not sure if you have included it in your simulations. I think you should clarify this point because in the discussion you describe the differences.
Table 2: Although the complete description of vegetation types can be consulted in ECOCLIMAP-II reference, it is not clear what “Oural” is. You should include a short description of each vegetation type, especially about its height.
191: Could you provide a deeper explanation about the pattern of effective roughness length for the multilayer SBL scheme? Why can we observe a flat variation for vegetation cover higher than 40%?
Figure 5: These figures don’t have axes, please add them
244: You have written “to between”
244: It is not necessary to write “degree”, write only 1.5 K.
246: Change insulation by insolation
241-242: When you analyze all the cases, you state different conclusions depending on the season and in the next section you analyze two cases. In the other events, are the meteorological conditions (wind, temperature, synoptic situation,…) similar to those analyzed? You could include some comments in the discussion.
Figure 8: It is not possible to derive the wind magnitude from the vector’s length if you don’t provide a scale.
272: I think that it would be interesting to show the evolution of cloud cover during the event, since in the previous section you related the impacts on ice forecast to the evolution of cloud cover (line 205).
Figure 9: Include in the figure caption a description of the red lines.
You should change the order of figures 10 and 11, since you comment them in this order.
305: Please, include the units of the liquid water content value shown.
Figures 11 and 15: The y-axis of figure b) is not clear.
330: You state that the timing of the onset varies by about 6h with higher vegetation fractions but Figure 13 does not show this difference, I can’t see more than 3 hours at the surface and at 117 m.
Round 2
Reviewer 1 Report
My comments were clearly addressed. Regarding the major comment 3, I meant how the four values of vegetation fractions with specific digits in Figure 10 were chosen? Maybe adding one sentence is enough. I recommend publication.
Author Response
Thank you again for your input. Figure 10 aims to give the reader an idea of how the cloud profile changed as the vegetation was increased from 0% percent vegetation to the reference case. Having figures for each of the 11 experiments seemed redundant as the qualitative change from one fraction to the next was generally continuous, so we opted to skip a few of the vegetation fractions for readability. We've added the sentence in the caption of Figure 10 to address this.